# Identification of Potential Phytochemical/Antimicrobial Agents against *Pseudoperonospora cubensis* Causing Downy Mildew in Cucumber through In-Silico Docking

**DOI:** 10.3390/plants12112202

**Published:** 2023-06-02

**Authors:** Nagaraju Jhansirani, Venkatappa Devappa, Chittarada Gopal Sangeetha, Shankarappa Sridhara, Kodegandlu Subbanna Shankarappa, Mooventhiran Mohanraj

**Affiliations:** 1Department of Plant Pathology, College of Horticulture—Bengaluru, University of Horticultural Sciences, Bagalkot 560 065, India; 2Center for Climate Resilient Agriculture, Keladi Shivappa Nayaka University of Agricultural and Horticultural Sciences, Shivamogga 577 201, India

**Keywords:** phytochemicals, antimicrobial compounds, homology modeling, molecular docking

## Abstract

Compatibility interactions between the host and the fungal proteins are necessary to successfully establish a disease in plants by fungi or other diseases. Photochemical and antimicrobial substances are generally known to increase plant resilience, which is essential for eradicating fungus infections. Through homology modeling and in silico docking analysis, we assessed 50 phytochemicals from cucumber (*Cucumis sativus*), 15 antimicrobial compounds from botanical sources, and six compounds from chemical sources against two proteins of *Pseudoperonospora cubensis* linked to cucumber downy mildew. Alpha and beta sheets made up the 3D structures of the two protein models. According to Ramachandran plot analysis, the QNE 4 effector protein model was considered high quality because it had 86.8% of its residues in the preferred region. The results of the molecular docking analysis showed that the QNE4 and cytochrome oxidase subunit 1 proteins of *P. cubensis* showed good binding affinities with glucosyl flavones, terpenoids and flavonoids from phytochemicals, antimicrobial compounds from botanicals (garlic and clove), and chemically synthesized compounds, indicating the potential for antifungal activity.

## 1. Introduction

The cucumber crop is widely grown in temperate and tropical regions of the world. It stands in fourth position after tomato (*Lycopersicon esculentum* Mill.), cabbage (*Brassica oleracea* var. *capitata* L.) and onion (*Allium cepa* L.). Cucumber has been considered an essential food source for over 5000 years and is used in culinary and non-culinary products. Fresh fruits are used in salads, pickles, cakes, and cooking. At the same time, processed cucumbers are used in sandwiches. Based on usage, cucumber fruits are divided into two types. “Pickling cucumbers” are mainly used in processing foods such as pickles. “Slicing cucumbers” are used for fresh consumption. Cucumbers are widely used as edible fruits because fruits are crispy, delicious, low in calories, rich in nutrients, and an excellent source of fiber needed for a healthy digestive system. The fruits of cucumbers possess several medicinal properties, namely, preventing constipation, having a cooling effect, and checking jaundice and indigestion [1,2,3,4]. Along with these, the consumption of cucumbers also provides good nutritional benefits to human beings. Every 100 g of cucumber fruit contributes 5 g of carbohydrates, 0.4 g of protein, 0.1 g of fat, 0.3 g of minerals, 10 mg of calcium, 0.4 g of fiber, and traces of vitamin C and iron. Cucumbers are a boon to the cosmetic industry. Many cosmetic products contain cucumber extracts, such as soaps, lotions, creams, and perfumes. In addition, the seeds of cucumbers are used in Ayurvedic preparations [5].

At the global level, about 397 million tons of cucumber were produced from 2,261,318 hectares of land, with average productivity of approximately 19.58 t/ha [6]. In India, 105 metric tons of cucumber was produced from an area of 1673 hectares [7]. Cucumbers are cultivated in several parts of India (Uttar Pradesh, Punjab, Rajasthan, Karnataka, and Andhra Pradesh). Cucumber is prone to several diseases like downy mildew, powdery mildew, fungal and bacterial wilts, and viral infections (cucumber mosaic virus, watermelon bud necrosis virus). It causes more economic losses with regard to production and export. Among these, downy mildew is a primary foliar disease that causes more damage and devastating losses to cucumber production [8].

Fungal diseases affect the quality and yield of crops. As one of the agricultural-limiting diseases, downy mildew on cucumber caused by *P. cubensis* significantly affects cucumber production. Cucumber downy mildew is reported to be found in more than 70 countries around the world. Cucumber downy mildew reduces cucumber yield by 10–20%, or even as much as 40%, without adequate control [9]. Management of *P. cubensis* is challenging because it can overcome the control measures (resistance and fungicide application) very quickly and has long-distance dispersal capacity. More usage of fungicides creates environmental pollution and health hazards. Usually, plants produce primary (proteins and polysaccharides) and secondary metabolites (alkaloids and flavonoids) that play an essential role in defense mechanisms. Phytochemical and antimicrobial compounds are known to boost resistance in plants [10]. Antimicrobial compounds and phytochemicals boost plant defenses by neutralizing fungal effector proteins [11]. Nowadays, researchers are focusing on finding potential phytochemicals or antimicrobial compounds against many plant diseases.

The effector proteins manipulate the structure, signaling, and metabolism of the host plant. Oomycetes produce effector proteins and virulence genes for pathogenesis [9]. Recent studies on the genome sequencing of *P. cubensis* and in silico analysis identified the effector proteins which play a role in the pathogenicity or virulence of *P. cubensis* infection. The genome sequencing of *P. cubensis* revealed the presence of 61 effector proteins with sequence similarity to the RXLR motif. The RXLR motif is an effector identified in the oomycetes of *P. cubensis*, the QXLR motif contains an effector designated as QNE. This effector protein plays a major role in the pathogenicity of *P. cubensis*. Genome sequencing of *Pythium insidiosum* revealed the involvement of four genes in pathogenesis viz., Exo-1, 3-beta glucanase, chitin synthase, and cytochrome oxidase subunit 1 [12].

Botanicals have anti-microbial properties and are used against many pathogens, including plant-pathogenic fungi and bacteria. The active compounds or chemical constituents of the botanicals act against pathogens. The botanicals used in this study, i.e., neem, tulsi, pudina, clove, and garlic are good sources of anti-microbial compounds and are used against many fungal pathogens, especially the oomycetes of fungi [13,14,15,16,17]. Binding interactions between two proteins of *P. cubensis* and ligands derived from *C. sativus* (L.), *Syzygium aromaticum* (L.) Merr. and L.M. Perry, *Ocimum tenuiflorum* (L.), *Allium cepa* (L.), *Mentha arvensis* (L.), and *Azadirachta indica* Juss, and fungicides viz., azoxystrobin, ridomil, kresoxim methyl, curzate and SAR inducers oxalic acid and salicylic acid were studied. Afterwards, molecular docking was carried out using 71 ligands (50 compounds from phytochemicals, 15 antimicrobial compounds, four fungicides, and two SAR inducers) with proteins as receptor targets.

The present study focused on the potentiality of phytochemicals present in *C. sativus* and antimicrobial compounds present in different botanicals which are easily available in the area of research conducted, namely garlic, clove, tulsi, mentha and neem, and chemically synthesized compounds against two proteins of *P. cubensis* associated with downy mildew of cucumber through homology modeling, in silico docking, and *in vitro* evaluation of botanicals against *P. cubensis*.

## 2. Material and Methods

### 2.1. Homology Modeling

The protein sequences of *P. cubensis* were downloaded from NCBI using accession numbers (Table 1). The protein modeling for protein sequences was carried out by using SWISS-MODEL (https://swissmodel.expasy.org) (accessed on 25 March 2021) [18] and the I-TASSER server. The templates were selected from the template identification wizard of SWISS-MODEL and later models were built. The output file was obtained in a PDB format that was used to visualize the model in PyMOL version 2.3 [19].

The SAVES-Procheck server (https://servicesn.mbi.ucla.edu/SAVES) (accessed on 28 March 2021) [20] was used to evaluate model quality with Procheck, errat, and verified by 3D Qmean plot. Then, the Ramachandran plot was obtained by Procheck in order to evaluate the model. ProtParam from the EXPASY server (www.expasy.ch/tools) (accessed on 28 March 2021) was [21] used to obtain the physicochemical properties of proteins like theoretical Isoelectric Point (PI), molecular mass, amino acid composition, atomic composition, extinction coefficient, instability index, estimated half-life and aliphatic index.

**Table 1 plants-12-02202-t001:** Protein sequences retrieved from National Centre for Biotechnology Information (NCBI).

Sl. No.	Sequence Description	Length of Proteins	Sequence of Amino Acids
1	Cytochrome oxidase subunit 1 of *P. cubensis* (Accession No. AEA38564.1)	412	MNFQNIKNWSTRWLFSTNHKDIGTXYLIFSAFAGIVGTTLSILIRIELAQPGNQIFMGNHQLYNVVVTAHAFVMVFFLVMPALIGGFGNWFVPLMIGAPDMAFPRMNNISFWLLPPALLLLISSAIVESGAGTGWAVYPPLSSVQAHSGPSVDLAIFSLHLTGISSLLGAINFISTIYNMRAPGLSFHRLPLFVWSILITAFLLLLTLPVLAGAITMLLTDRNLNTSFYDPSGGGDPVLYQHLFWFFGHPEVYVLILPAFGIISQVSAYFAKKNVFGYLGMVYAMLSIGLLGSIVWAHHMFTVGLDVDTRAYFSAATMIIAVPTGIKIFSWLATLWGGSLKFETPLLFTLGFILLFVMGGVTGVVMSNSGLDIALHDTYYIVGHFHYVLSMGAIFGIFTGFYFWIGKISGRR
2	QNE 4 effector protein *P. cubensis* (Accession No. ADW27474.1)	517	MMPPAKLVAYIAVASSIVLARYEASTDITSTSDANKLSISAPSDPVQHDTKQLLRTSDTAVTKDNEERMFNAAGLKRASTMSHFADVHGLPHEPLAPHLHDTYDPAGASHPPVLPYTGEAKAHEDLQHAASTSNPLKKISPADTQLTEGENNEAEILKRIMTLMQPVAPRALKRKRKLPDGTETQLQWNESDILDIYEKHKDKFLNIMNEWWLNGLGPQAFERMILENQLPTSIYEDYVMFHAAKDEEMYEHFAKWQNEGILPKEIEEKINAVLPKARKAPLVVRLENKYEVFYKKKQPFEAYRTKLLDEDTEPEEAERLKSKKWDRLRVVLKVRSSQRKTKFTLQWFRKHPNEFLLKSIQEGTPPEDIRSVLGLARLEGLKLFKHPNYEYYLKYLKLWFQTHSTEHWQERVPKGMPPEDVRFILGLGQLKGSEFSQHPNFPEYIKFFELWHEAYTRKKMKEWMQLNTPLDEAFAKLAIRDHNDVEFIVDKSDLYMKQYENEWKKKHPTLRTPAVST

### 2.2. Molecular Docking 

#### 2.2.1. Ligands’ Source and Fungal Receptor Proteins

The phytochemicals present in *C. sativus*, antimicrobial compounds from botanicals viz., *Ocimum tenuiflorum*, *Allium cepa*, *Syzygium aromaticum*, *Azadirachta indica*, and *Mentha arvensis*, and fungicides were obtained from the published literature [22,23,24,25,26,27,28]. A total of 71 compounds were selected for molecular docking, details of these compounds are given in Table 2. The three-dimensional (3D) structures of proteins (QNE and cytochrome oxidase subunit 1) were obtained from the protein data bank (www.rcsb.org) (accessed on 25 March 2021). Similarly, 3D confirmers of the selected ligands were retrieved from the PubChem (https://pubchem.ncbi.nlm.nih.gov) (accessed on 25 March 2021 database in PDB and SDF formats, respectively.

#### 2.2.2. Preparation of Ligands and Target Proteins

Using Avogadro version 1.2.0 [29] with force field type MMFF94, the ligands’ 3D structures were optimized and then translated to PDB format using Open Babel version 3.1.1. Further simplification was attained by running the optimized ligands with the lowest energy through the AutoDock-MGL tools [30], adding the Gasteiger charges, and obtaining the PDBQT files via standard processes. A PyMOL check of the downloaded 3D structures was made to check for side-chain anomalies, improper bonds, and missing hydrogens [19]. Using Biovia Discovery Studio 2020, all water molecules, ions, complex molecules of ligands, and proteins were removed [31]. A PDB structure was optimized with Auto Dock-MGL by adding the polar hydrogens to obtain the PDBQT files.

#### 2.2.3. Active Site Prediction and Molecular Docking

Using Biovia Discovery Studio 2020, the active sites of fungal proteins were determined. Molecular docking of optimized ligands and proteins in PDBQT format was performed using Auto Dock Vina software [30]. Auto Dock Vina software uses its scoring function (binding affinity) to predict the interaction between ligand and protein. A grid box of 60 Å × 60 Å × 60 Å was used for proteins with different XYZ coordinates based on predicted active sites for molecular docking. After docking analysis, the output file consists of the top nine binding poses, with their respective binding affinity in kcal/mol. The ligand binding poses with the highest binding affinity and the lowest root mean square deviation (RMSD) were chosen. The protein-ligand interaction in 3D structure was visualized in PyMOL. The two-dimensional (2D) structure was also visualized in Biovia Discovery Studio 2020. The 3D visualization indicates the target protein’s binding pocket or precise location.

On the other hand, the 2D structure visualization shows the different bonds formed between the amino acid residues of the fungal target protein and ligand. The workflow of molecular docking of compounds with proteins of *P. cubensis* associated with cucumber is depicted in Figure 1. The botanicals studied in molecular docking were further evaluated under in vitro conditions.

### 2.3. In Vitro Evaluation of Botanicals

The botanicals were tested at three different concentrations of 5, 10, and 15% by m/v. The required concentration of botanicals was extracted by two different solvents.

#### 2.3.1. Aqueous Extraction

Leaf samples from neem, tulsi, pudina, and cloves of garlic and clove were collected from the fields, College of Horticulture, Bengaluru, India. A hundred grams of each botanical sample were cleaned with tap water and shade dried at room temperature until complete evaporation of moisture. The samples were then made into powder by using an electric blender. Three concentrations of 5, 10, and 15% were prepared by suspending 5 g, 10 g, and 15 g of each botanical powder in 100 mL of sterile distilled water followed by filtration+ through cheesecloth to remove unwanted coarse particles. The filtered extract was centrifuged at 5000 rpm for 5 min to obtain a clear extract [32,33,34,35].

#### 2.3.2. Methanolic Extraction

The procedure for the methanolic extraction of the botanicals was followed according to [35]. Leaf samples from neem, tulsi, and pudina, and cloves of garlic and clove were collected from the college farm located in Bengaluru, India. A hundred grams of each botanical sample were cleaned and made into powder. Thirty grams of each powdered botanical were extracted with 90 mL of methanol and kept on a rotary shaker for three days with periodic shaking. Then, the extract was filtered with muslin cloth and centrifuged at 5000 rpm for 15 min. The supernatant was collected in tubes and kept in a hot air oven until complete evaporation of the solvent. Then the leftover material in the tubes was utilized for the experimentation.

The fresh sporangia of *P. cubensis* were collected from the naturally infected cucumber research plot located at the College of Horticulture, Bengaluru, India. The procedure for sporangia collection was followed as per Bommesh et al. [34]. Five-day-infected cucumber leaves were picked and cut into small pieces before being soaked in sterile distilled water to make a sporangial suspension. Using a hemocytometer, the sporangia concentration was adjusted to 100 sporangia/mL. Then, a drop of sporangia suspension was mixed with a drop of botanical extract of 5%, 10%, and 15%, respectively, and kept in a BOD incubator at 20 °C and 100 percent relative humidity for 2 h. After 2 h of incubation, the sporangial germination was recorded under a microscope. A cavity slide with sterile distilled water was maintained as the control. The percentage of sporangia germination was calculated by the given formula.
Percent Germination of sporangia (PG) = (A/B) × 100
where,
A = Number of sporangia germinated
B = Number of sporangia observed

The percent inhibition was calculated by the given formula
Percent inhibition of sporangial germination = (C − T)/C × 100
where,
C = Germination of sporangia in control
T = Germination of sporangia in treatment.

The experiment was laid out with a Completely Randomized Design (CRD) with three replications. The cavity slides of each botanical concentration (5%, 10%, and 15%) were maintained with three replications along with the control under similar conditions. All slides were kept in a BOD incubator at 20 °C and 100 percent relative humidity for 2 h. The percentage of sporangia was calculated from all three replications along with the control, then analysis of variance was performed from the mean values,. The data were changed into arc-sine transformation for statistical analysis using OPSTAT [36].

## 3. Results

### 3.1. Modeling and Physicochemical Properties of Proteins

#### 3.1.1. Prediction of the 3D Structure of Proteins of *P. cubensis*

The two protein sequences of *P. cubensis* were obtained and annotated (Table 1). The BLASTn results showed high query coverage (>99%) and percent identity (>99.47%) in both the proteins of *P. cubensis*. Later, these sequences were selected for protein modeling using SWISS-MODEL.

#### 3.1.2. Template Selection

The selection of templates for building homology models was performed using the wizard of SWISS-MODEL with the following criteria: the template should show high coverage, i.e., >65 percent of the target aligned to the template and sequence identity should be more than 30 percent. Then, we used the GMQE and QMEAN scoring functions as initial criteria to discriminate good models from bad. Higher GMQE and QMEAN scores and acceptable alignment values were obtained during modeling, suggesting that statistically acceptable homology models were generated [37]. The output file was obtained in a PDB format that was used to visualize the model in PyMOL version 2.3. [19]. Global model quality estimation (GMQE) is the quality estimation that combines properties from the target-template alignment. The quality estimate ranges between 0 and 1 with higher values for better models. Qualitative model energy analysis (QMEAN) is a composite scoring function describing the major geometrical aspects of protein structures (Table 3).

The results showed that the predicted cytochrome oxidase subunit 1 protein of *P. cubensis* model had 44.77 percent alpha-helices with beta turns comprising 8.27 percent, whereas the QNE4 effector protein has 42.36 percent alpha-helices with 8.70 percent beta turns (Table 4).

#### 3.1.3. Ramachandran Plot Analysis

The Ramachandran plot indicated the phi-psi torsion angle for all residues in the structure (except those at the chain termination). The darkest areas correspond to the ‘core’ region representing the most favorable combinations of phi-psi values. Ideally, one would hope to have over 90 percent of the residues in these ‘core’ regions. The percentage of residues in the ‘core’ region is one of the best guides to stereo-chemical quality. A good quality Ramachandran plot has over 90 percent in the most favored region [38].

Ramachandran plot analysis was carried out for two proteins (cytochrome oxidase subunit 1 and QNE4) of *P. cubensis*. The QNE4 effector protein was shown to have 86.8 percent of residues in the favored region (red color), 12.3 percent in the additionally allowed area (yellow color), 0 percent of residues in the generously allowed region (beige color), and 0.9 percent of residues in the disallowed region (white color) (Figure 2a). Similarly, the cytochrome oxidase subunit 1 protein had 82.8 percent of residues in the favored region (red color), 16.0 percent in the additionally allowed region (yellow color), 1.2 percent of residues in the generously allowed region (beige color), and 0 percent of residues in the disallowed region (white color) (Table 5) (Figure 2b). Homology modeling plays a vital role in structural proteomics and developing or designing potential compounds using an in silico approach.

#### 3.1.4. Physico-Chemical Properties of Two Proteins of *P. cubensis*

The physico-chemical properties of proteins of *P. cubensis* were determined by ProtParam from the EXPASY server (www.expasy.ch/tools) (accessed on 28 March 2021) [21] and furnished in Table 6. The extinction coefficient indicates how much light a protein absorbs at a particular wavelength. The instability index estimates the protein’s stability in a test tube. If it is greater than 40, it is not stable; hence the effector QNE4 protein was stable in nature and another protein, cytochrome oxidase subunit 1, was unstable in nature. The grand average of hydropathic (GRAVY) value, which is calculated as the sum of the hydropathic values of all the amino acids divided by the number of residues in the sequence. A negative GRAVY value indicates that the protein is non-polar and a positive value indicates that the protein is polar. Hence, our results revealed that both proteins are non-polar in nature (Table 6). The overall stereochemical properties of the generated models were highly reliable and valuable in understanding the protein function.

**Figure 2 plants-12-02202-f002:**
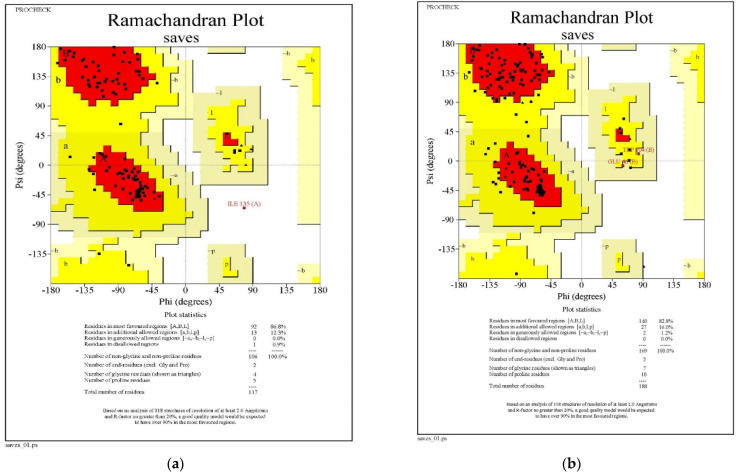
Comparative protein model quality assessment by using a Ramachandran plot for (**a**) QNE4 and (**b**) cytochrome oxidase subunit 1 proteins.

**Table 5 plants-12-02202-t005:** Ramachandran plot statistics for QNE4 and cytochrome oxidase subunit 1 proteins.

Sl. No.	Ramachandran Plot Statistics	QNE4		Cytochrome Oxidase Subunit 1	
		Residues	Percentage (%)	Residues	Percentage (%)
1	Residues in most favored regions [A, B, L]	92	86.8	140	82.8
2	Residues in additional allowed regions [a, b, l, p]	13	12.3	27	16.0
3	Residues in generously allowed regions [~a,~b,~l,~p]	0	0.0	2	1.2
4	Residues in disallowed regions	1	0.9	0	0.0
5	Number of non-glycine and non-proline residues	106	100.0	169	100.0
6	Number of end-residues (except Gly and Pro)	2		2	
7	Number of glycine residues (shown in triangles)	4		7	
8	Number of proline residues	5		10	
9	Total number of residues	117		188	

**Table 6 plants-12-02202-t006:** Physico-chemical parameters computed using Expasy’s ProtParam tool.

Sl. No	Description	QNE4	Cytochrome Oxidase Subunit 1
1	Number of amino acids	517	412
2	Molecular weight (Daltons)	60,203.75	45,283.65
3	Theoretical pI	7.08	8.70
4	Negatively charged residues	76	14
5	Positively charged residues	75	16
6	Ext. coefficient M^−1^ cm^−1^	88,810	82,850
7	Instability index	40.53	25.94
8	Aliphatic index	75.71	114.78
9	Grand average of hydropathicity (GRAVY)	−0.726	−0.796

### 3.2. Molecular Docking Studies

To develop effective phytochemicals/antimicrobial compounds from botanicals against *P. cubensis* associated with cucumber, approximately 71 compounds from plant and chemical sources were used for molecular docking with proteins as a potential target. Before the docking analysis, the ligands were optimized by minimizing the energy with force field type MMFF94, and this helps in removing clashes among atoms and developing a stable starting pose of the ligands for binding interaction [39]. The docking, coupled with a scoring function, can be utilized to screen a large number of potential phytochemicals in silico. Generally, in molecular docking, a binding affinity lower than the upper threshold (−6 kcal/mol) is considered the cut-off value for concluding good binding affinity between protein and ligand [39]. The 3D and 2D visualization of phytochemicals, antimicrobial compounds, and chemically synthesized compounds based on binding affinity with respective fungal receptor proteins has been represented (Appendix A), (Figure 3, Figure 4, Figure 5 and Figure 6). Hydrogen bond energy majorly contributed to the score [40] of selected compounds used in the current molecular docking studies against two proteins of *P. cubensis*, which displayed very good dock scores above the threshold cut-off of −6 kcal/mol (Table 7). The ligand structures and necessary hydrogen bond formation between the top phytochemicals, antimicrobial compounds, and fungicides with their respective fungal protein receptors have been illustrated in Table 8, Table 9, Table 10 and Table 11.

**Table 7 plants-12-02202-t007:** Dock score of interactions between phytochemicals, antimicrobial compounds, botanicals, and chemically synthesized compounds against cytochrome oxidase subunit 1 and QNE4 effector protein of *P. cubensis*.

Group	Sl. No.	Compound	Dock Score for Binding Affinity (kcal/mol)	
			Cytochrome Oxidase Subunit 1	QNE Effector Protein *P. cubensis*
Terpenoids	1	Cucurbitacin-A	−7.9	−8.1
	2	Cucurbitacin-B	−7.8	−8.3
	3	Cucurbitacin-C	−7.4	−7.4
	4	Cucurbitacin-D	−8.0	−8.2
	5	Cucurbitacin-E	−8.0	−8.1
	6	Cucurbitacin-I	−8.3	−8.0
Glucosyl flavones	7	Cucumerin-A	−7.8	−9.1
	8	Cucumerin-B	−7.7	−8.5
Flavonoids	9	Vitexin	−7.0	−7.5
	10	Isovitexin	−7.6	−8.0
	11	Orientin	−7.1	−7.4
	12	Isoorientin	−7.4	−7.9
Megastigmane derivatives	13	Cucumegastigmane-I	−5.3	−5.4
	14	Cucumegastigmane-II	−6.2	−7.8
	15	(+)-Dehydrovomifoliol	−5.0	−6.3
Indolic secondary metabolites	16	Indole-3-aldehyde	−4.4	−5.0
	17	Indole-3-carboxylic acid	−4.8	−5.5
Flavone glucosides	18	Isoscoparin	−7.5	−8.5
	19	Saponarin	−8.1	−7.3
	20	Vicenin-2	−7.1	−8.0
	21	Apigenin-7-*O*-glucoside	−7.5	−8.5
	22	Quercetin-3-*O*-glucoside	−6.8	−7.6
	23	Isorhamnetin-3-*O*-glucoside	−6.5	−6.9
	24	Kaemferol-3-*O*-rhamnoside	−7.2	−7.4
Polyphenol	25	4-hydroxycinnamic acid	−4.8	−5.8
Antimicrobial compounds	26	Carrageenan	−7.0	−8.0
	27	Acyclovir	−5.0	−5.7
	28	5-Azacytidine	−5.4	−5.8
	29	Cytarbine	−5.4	−5.9
	30	Ribavirin	−5.2	−5.8
	31	Ridovudine	−6.0	−7.0
	32	Ningnanmycin	−6.0	−7.8
	33	Vidarabine	−5.7	−6.6
	34	Acycloguanosine	−5.0	−5.5
	35	2-Thiouracil	−3.3	−4.6
	36	Moroxydine hydrochloride	−4.7	−5.2
	37	Luotonin A	−7.8	−6.9
	38	Tylophorinine	−7.1	−8.1
	39	Antofine	−7.2	−6.6
	40	Deoxytylophorinine	−7.2	−6.7
	41	Pyrroloisoquinoline	−5.0	−7.4
	42	Pulmonarin-A	−4.8	−5.5
	43	Pulmonarin-B	−5.0	−5.9
	44	Streptindole	−6.1	−7.6
	45	Tryptanthrin	−6.8	−7.7
	46	Essramycin	−6.6	−7.9
	47	Chlorogenic acid	−6.7	−7.3
	48	Peonidin	−6.3	−7.2
	49	Swertianolin	−8.0	−7.8
	50	Zidovudine	−5.8	−6.6
Clove	51	Eugenol	−4.7	−5.3
	52	Eugenol acetate	−4.7	−5.4
	53	(E)-β-caryophyllene	−5.6	−6.8
Garlic	54	Allyl acetate	−6.6	−7.2
	55	Allicin	−3.4	−3.7
	56	Allixin	−4.9	− 5.8
	57	Alliin	−3.9	−4.5
Neemm	58	Azadiractin a	−3.7	3.5
	59	Nibolin b	−3.4	−3.4
	60	Azadiractin b	−3.2	−3.6
	61	Nimbin	−5.0	−5.5
Tulsi	62	Gallic acid	−4.5	−5.1
	63	Catechol	−4.1	−5.0
	64	Cinnamic acid	−3.7	−4.4
Pudina	65	Menthol	−3.3	−4.8
Fungicides	66	Azoxystrobin	−7.2	−8.1
	67	Ridomil	−5.3	−5.3
	68	Kresoxim methyl	−6.3	−4.4
	69	Curzate	−5.3	−6.0
	70	Oxalic acid	−3.3	−5.3
	71	Salicylic acid	−4.5	−6.5

**Table 8 plants-12-02202-t008:** Number of hydrogen bonds formed during the interaction between top phytochemicals/antimicrobial compound structures with the QNE 4 effector protein of *P. cubensis* associated with cucumber.

Sl. No.	Compound with PubChem ID	Structural and Chemical Formula	No. of H Bonds	Amino Acid Residue of QNE 4 Effector Protein Involved in Hydrogen Bonding with Ligand
1	Cucumerin-A 44257649	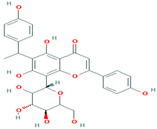 C_29_H_28_O_11_	4	ARG339, TVR290, LEU 126, ASN134
2.	Cucumerin-B 44257648	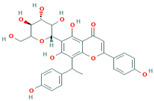 C_29_H_28_O_11_	1	HIS110
3	Isoscoparin 442611	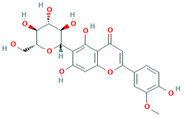 C_22_H_22_O_11_	3	SER109, HS110, GLY217
4	Apigenin-7-*O*-glucoside 5280746	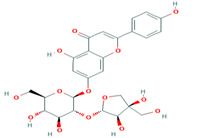 C_26_H_28_O_14_	4	ASN214, SER109, MET224, GLY107
5	Cucurbitacin-B 5281316	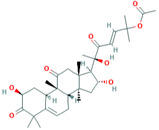 C_32_H_46_O_8_	1	HIS110
6	Cucurbitacin-D 5281318	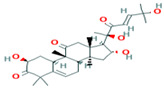 C_32_H_44_O_7_	3	SER 125, TYR108,ARG146
7	Cucurbitacin-A 5281315	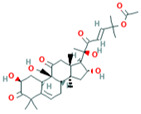 C_32_H_46_O_9_	2	SER140, SER82
8	Cucurbitacin-E 5281319	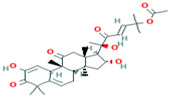 C_32_H_44_O_8_	2	SER 82, SER 109
9	Cucurbitacin-I 5281321	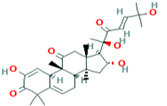 C_30_H_42_O_7_	3	LYS121,GLS127,ARG339
10	Vicenin-2 442664	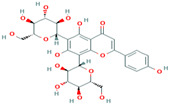 C_27_H_30_O_15_	4	SER109, HIS83, GLY107, SER82

**Table 9 plants-12-02202-t009:** The number of hydrogen bonds formed during the interaction between top antimicrobial compounds from botanicals and chemically synthesized compound structures with the QNE4 effector protein of *P. cubensis* associated with cucumber.

Sl. No.	Compound with PubChem ID	Structural and Chemical Formula	No. of H Bonds	Amino Acid Residue of QNE4 Effector Protein Involved in Hydrogen Bonding with Ligand
1	Azoxystrobin 3034285	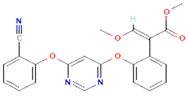 C_22_H_17_N_3_O_5_	1	SER109
2	Allyl acetate 11584	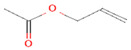 C_5_H_8_O_2_	2	ASP86, HIS83
3	Salicylic acid 338	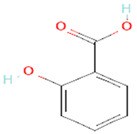 C_7_H_6_O_3_	3	ALA376,ARG377,LEU381
4	Curzate 5364079	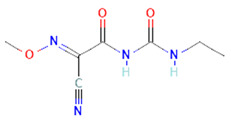 C_7_H_10_N_4_O_3_	4	THR456,ALA376
5	Allixin 86374	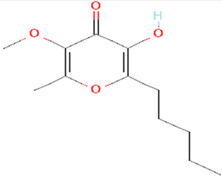 C_12_H_18_O_4_	2	ARG285,GLN165

**Figure 3 plants-12-02202-f003:**
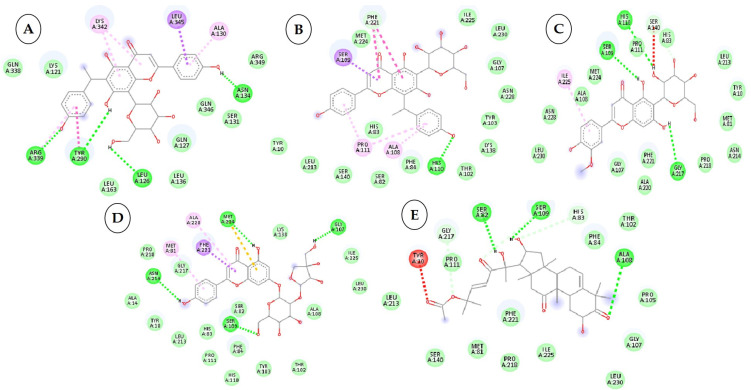
Two-dimensional visualization of the interaction between the QNE 4 effector protein and the top five phytochemicals (**A**) Cucumerin A (**B**) Cucumerin B (**C**) Isocarpin (**D**) Apigenin-7-*O-* glucoside (**E**) Cucurbitacin-B.

**Figure 4 plants-12-02202-f004:**
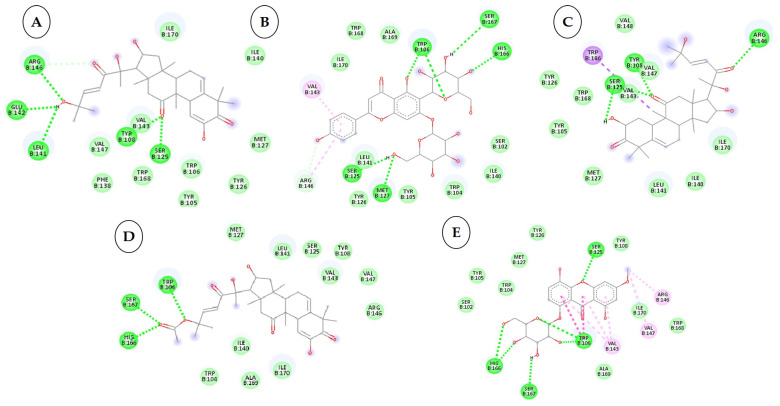
Two-dimensional visualization of the interaction between the cytochrome oxidase subunit 1 protein and the top five phytochemicals (**A**) Cucurbitacin-I (**B**) Saponarin (**C**) Cucurbitacin-D (**D**) Cucurbitacin-E (**E**) Swertianolin S.

**Figure 5 plants-12-02202-f005:**
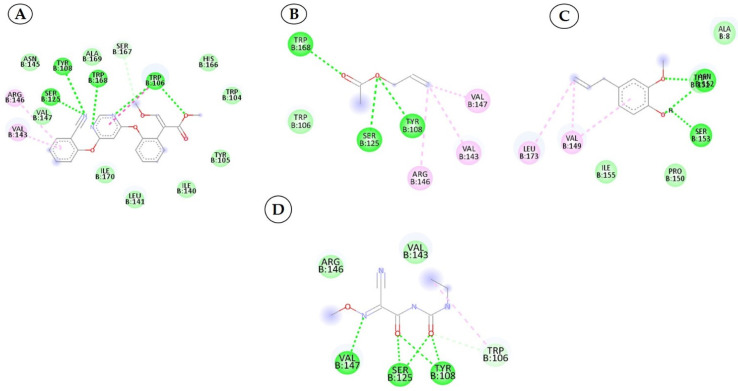
Two-dimensional visualization of the interaction between the cytochrome oxidase subunit 1 protein and top compounds from botanicals and chemical sources (**A**) Azoxystrobin (**B**) Allyl acetate (**C**) Kresoxim methyl (**D**) Curzate.

**Figure 6 plants-12-02202-f006:**
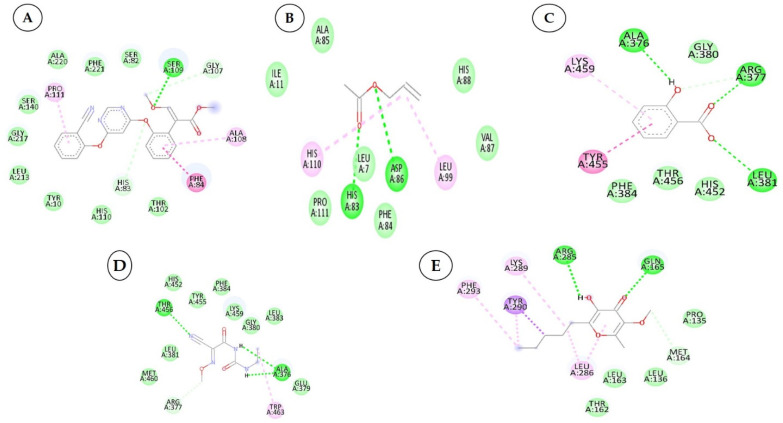
Two-dimensional visualization of the interaction between the QNE 4 effector protein and top compounds from botanicals and chemical sources (**A**) Azoxystrobin (**B**) Allyl acetate (**C**) Salicylic acid (**D**) Curzate (**E**) Allixin.

### 3.3. Interactions between the QNE4 Effector Protein and Phytochemicals, Antimicrobial Compounds, and Chemically Synthesized Compounds

Molecular docking analysis of QNE 4 with 50 phytochemicals showed that the majority of the compounds bind to the effector protein of *P. cubensis* with favorable binding energies ranging from −4.4 kcal/mol (for Indole-3-aldehyde) to −9.1 kcal/mol (cucumerin-A), whereas antimicrobial compounds from different botanical sources and fungicides showed binding energies in the range of −3.4 to −12.1 (Table 7). Among the 50 phytochemicals, cucumerin-A (−9.1 kcal/mol), Isocarpin (−8.5 kcal/mol), apigenin −7-*O*-glucoside (−8.5 kcal/mol), cucumerin-B (−8.5 kcal/mol), cucurbitacin-B (−8.3 kcal/mol), cucurbitacin-D (−8.2 kcal/mol), cucurbitacin-A and cucurbitacin-E (−8.1 kcal/mol), cucurbitacin-I (8.0 kcal/mol), vincein (−8.0 kcal/mol), and caragenin (−8.0 kcal/mol) were the top 10 compounds with the highest binding affinities. The phytochemical compounds belonging to glucosyl flavones, terpenoids, and flavonoids have shown an excellent inhibitory action on the ONE4 effector protein of *P. cubensis.* Among the 15 antimicrobial compounds from botanicals tested, azoxystrobin (−8.1 kcal/mol), allyl acetate (−7.2 kcal/mol), (E)-β-caryophyllene (−6.8 kcal/mol), salicylic acid (−6.5 kcal/mol), curzate (−6.0 kcal/mol), and allixin (−5.8 kcal/mol) showed highest binding affinities (Table 7). The antimicrobial compounds obtained from botanicals namely, garlic and clove have shown a good inhibitory action on ONE4 effector protein of *P. cubensis*. At the same time, azoxystrobin (−8.1 kcal/mol), salicylic acid (−6.5 kcal/mol) and curzate (−6.0 kcal/mol) are the chemical compounds which exhibited the highest binding affinities. Overall, cucumerin-A (−9.1 kcal/mol) showed good inhibitory action on the ONE4 effector protein of *P. cubensis* out of 71 compounds tested.

Among the phytochemical compounds, cucumerin-A (−9.1 kcal/mol) exhibited the highest docking score with the QNE 4 effector protein. The ARG339, TVR290, LEU126, ASN134 amino acid residue is involved in forming four hydrogen bonds in the binding pocket of the QNE 4 effector protein. Similarly, cucumerin-B interacted with the HS110 amino acid residue by forming one hydrogen bond. Likewise, isoscoparin interacted with the SER109, HS110, and GLY217 amino acid residues by forming three hydrogen bonds, apigenin −7-*O*-glucoside showed an interaction with the ASN214, SER109, MET224, and GLY107 amino acids and produced four hydrogen bonds, the HIS110 amino acid shared one hydrogen bond with cucurbitacin-B, three hydrogen bonds of the SER82, SER109, and ALA108 amino acids were generated upon interaction with cucumerin-B, the SER82, SER109, and ALA108 amino acids of cucurbitacin-D were involved in forming three hydrogen bonds, the SER140 and SER82 amino acids of cucurbitacin-A interacted with two hydrogen bonds, the LYS121, GLN127, and ARG339 amino acids of cucurbitacin-I contributed three hydrogen bonds, vicenin-2 created an interaction with the SER109, HIS83, GLY107, and SER82 amino acids and generated four hydrogen bonds, and carrageenan interacted with the SER109, PHE84, HIS83, and HIS110 amino acids by forming four hydrogen bonds with the binding of the QNE4 effector protein of *P. cubensis* (Table 8).

In binding interactions between 15 antimicrobial compounds from different botanicals and six compounds from chemical sources and QNE 4, the docking score ranged from −3.4 to −8.1. Out of 21 compounds, the azoxystrobin (−8.1 kcal/mol) chemical compound showed the top docking score with the QNE 4 effector protein and interacted with SER109 amino acid residues to form one hydrogen bond in the binding pocket of the QNE 4 effector protein. Likely, allyl acetate created an interaction with the ASP86 and HIS83 amino acids and produced two hydrogen bonds; three hydrogen bonds of the ALA376, ARG377, and LEU381 amino acids were generated upon interaction with salicylic acid, the THR456 and ALA376 amino acids of curzate were involved in forming two hydrogen bonds, and the ARG285 and GLN165 amino acids shared two hydrogen bonds with allixin with the QNE4 effector protein of *P. cubensis* (Table 9).

### 3.4. Interactions between the Cytochrome Oxidase Subunit 1 Protein and Phytochemicals, Antimicrobial Compounds, and Fungicides

Among the 50 phytochemicals used for screening against the cytochrome oxidase subunit 1 protein, Indole-3-aldehyde has shown the lowest dock score of −4.4 kcal/mol and cucurbitacin-I have shown the highest dock score of −8.3 kcal/mol (Table 7). Ten compounds; cucurbitacin-I (−8.3 kcal/mol), saponarin (−8.1 kcal/mol), cucurbitacin-D (−8.0 kcal/mol), swertianolin (−8.0 kcal/mol), cucurbitacin-E (−8.0 kcal/mol), cucurbitacin-A (−7.9 kcal/mol), cucurbitacin-B (−7.8 kcal/mol), cucumerin-A (−7.8 kcal/mol), luotonin A (−7.8kcal/mol), and cucumerin-B (−7.7 kcal/mol) exhibited better dock scores. The phytochemicals from terpenoids, glucosyl flavones, and the flavone glucosides group have shown good affinities with the target cytochrome oxidase subunit 1 protein of *P. cubensis*.

Cucurbitacin-I interacted with the ARG461.GLU142, LEU141, TYR108, and SER125 amino acid residues through forming five hydrogen bonds with the cytochrome oxidase subunit 1 protein of *P. cubensis*. Likewise, the TRP106, SER167, HIS166, SER125, and MET127 amino acids of catechin shared five hydrogen bonds, cucurbitacin-D displayed an interaction with the ARG146, TYR108, and SER125 amino acids and produced three hydrogen bonds, three hydrogen bonds of the TRP106, SER167, and HIS166 amino acids were generated upon interactions with cucurbitacin-E, swertianolin created an interaction with the SER125, HIS166, SER167, and TRP106 amino acids and developed four hydrogen bonds, the TRP168, SER125, SER167, HIS166, and TRP104 amino acids of cucurbitacin-A were involved in forming five hydrogen bonds, cucurbitacin-B interacted with the ASN152, ARG146, LEU141, VAL145, and VAL147 amino acids by forming five hydrogen bonds, the LEU141, SER125, TRP106, and SER167 amino acids of cucumerin-A contributed four hydrogen bonds, Luotonin A interacted with the TYR108 and SER125 amino acids by forming two hydrogen bonds, and cucumerin-B interacted with the LEU141, MET127, SER125, TYR108, and SER167 amino acids by forming five hydrogen bonds with the active site of the cytochrome oxidase subunit 1 protein (Table 10).

The docking score for the 21 antimicrobial compounds and fungicides ranged from −3.2 kcal/mol (for azadiractin b) to −7.2 kcal/mol (for azoxystrobin) (Table 7). Four compounds; azoxystrobin (−7.2 kcal/mol), allyl acetate (−6.6 kcal/mol), kresoxim methyl (−6.3 kcal/mol), and curzate (−5.3 kcal/mol) exhibited uppermost binding affinities (Table 7). The compounds from chemical sources and antimicrobial compounds from garlic showed superior affinities with the target cytochrome oxidase subunit 1 protein of *P. cubensis*. Azoxystrobin interacted with the SER125, TYR108, TRP168, and TRP106 amino acid residues in forming four hydrogen bonds with the cytochrome oxidase subunit 1 protein of *P. cubensis*. Similarly, the SER125, TYR108, and TRP168 amino acids shared three hydrogen bonds with allyl acetate, and two hydrogen bonds of the TAM 552 and SER153 amino acids were interfaced with kresoxim methyl. The VAL147, SER125, and TYR108 amino acids of curzate contributed three hydrogen bonds with the active sites of the cytochrome oxidase subunit 1 protein of *P. cubensis* (Table 11).

**Table 10 plants-12-02202-t010:** Number of hydrogen bonds formed during the interactions between top phytochemicals/antimicrobial compound structures and the cytochrome oxidase subunit 1 protein of *P. cubensis* associated with cucumber.

Sl. No.	Compound with PubChem ID	Structural and Chemical Formula	No. of H Bonds	Amino Acid Residue of Cytochrome Oxidase Subunit 1 Protein Involved in Hydrogen Bonding with Ligand
1	Cucurbitacin-I 5281321	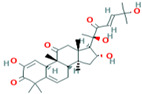 C_30_H_42_O_7_	5	ARG461.GLU142, LEU141,TYR108,SER125
2	Saponarin 441381	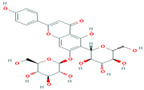 C_22_H_30_O_15_	5	TRP106, SER167,HIS166,SER125,MET127
3	Cucurbitacin-D 5281318	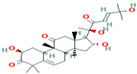 C_32_H_44_O_7_	3	ARG146, TYR108, SER125
4	Cucurbitacin-E 5281319	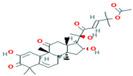 C_32_H_44_O_8_	3	TRP106, SER167, HIS166
5	Swertianolin 5858086	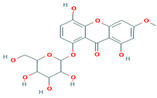 C_20_H_20_O_11_	4	SER125, HIS166, SER167, TRP106
6	Cucurbitacin-A 5281315	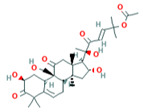 C_32_H_46_O_9_	5	TRP168, SER125, SER167, HIS166, TRP104
7	Cucurbitacin-B 5281316	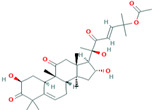 C_32_H_46_O_8_	5	ASN152, ARG146, LEU141, VAL145, VAL147
8	Cucumerin-A 44257649	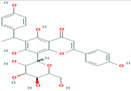 C_29_H_28_O_11_	4	LEU141, SER125, TRP106, SER167
9	Luotonin A 10334120	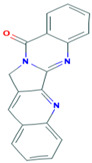 C_18_H_11_N_3_O	2	TYR108, SER125
10	Cucumerin-B 44257648	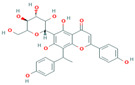 C_29_H_28_O_11_	5	LEU141, MET127, SER125, TYR108, SER167

**Table 11 plants-12-02202-t011:** Number of hydrogen bonds formed during the interaction between top antimicrobial compounds from botanicals and chemically synthesized compound structures and the cytochrome oxidase sub-unit 1 protein of *P. cubensis* associated with cucumber.

Sl. No.	Compound with PubChem ID	Structural and Chemical Formula	No. of H Bonds	Amino Acid Residue of Cytochrome Oxidase Subunit 1 Protein Involved in Hydrogen Bonding with Ligand
1	Azoxystrobin 3034285	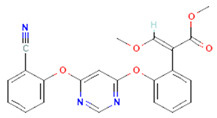 C_22_H_17_N_3_O_5_	4	SER125, TYR108, TRP168, TRP106
2	Allyl acetate 11584	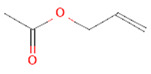 C_5_H_8_O_2_	3	SER125, TYR108 and TRP168
3	Kresoxim methyl 6112114	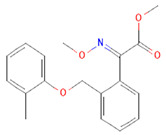 C_18_H_19_NO_4_	2	TAM 552, SER153
4	Curzate 5364079	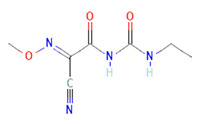 C_7_H_10_N_4_O_3_	3	VAL147, SER125, TYR108

### 3.5. In Vitro Evaluation of Botanicals

Evaluation of botanicals against sporangial germination of *P. cubensis* in vitro was carried out at different concentrations of five botanicals. The data revealed that all the treatments (botanicals) significantly inhibited the sporangial germination of *P. cubensis*. Among all of the botanicals tested, garlic bulb extract at 15 percent concentration showed significantly higher percentage inhibition (71.42%) followed by clove oil (64.51%) (Figure 7). The slightest inhibition of sporangial germination (33.33%) was observed at 5 percent concentration of neem (Table 12).

## 4. Discussion

In the present investigation, glucosyl flavones (cucumerin A, cucumerin B), terpenoids (cucurbitacin-A, cucurbitacin-B, cucurbitacin-C, cucurbitacin-D, cucurbitacin-E, and cucurbitacin-I), flavanone glucosides (isocarpin, apigenin-7-*O*-glucoside, vicenin-2, and saponarin), and antimicrobial compounds (luotionin) have shown good binding interactions on the ONE4 and cytochrome c oxidase subunit 1 proteins of *P. cubensis*. Similarly, luotonin-A has shown broad-spectrum fungicidal activities against 14 different phytopathogenic fungi [26].

Among the botanicals tested, antimicrobial compounds from garlic (allyl acetate, allicin, and alliin) and clove (eugenol acetate and (E)-β-caryophyllene) showed an excellent binding affinity with the ONE4 and cytochrome c oxidase subunit 1 proteins of *P. cubensis*. It was reported that the alliin from garlic showed significant binding interactions with the target-Avr3a11 effector protein of *Phytopthora capsici* compared to the commonly used fungicides, indicating that alliin can act as a potential inhibitor of Avr3a11 [40]. It was revealed that chemical compounds from garlic have antioxidant properties by conducting molecular docking analysis of the chemical compounds of garlic against NADPH oxidase [41]. The best docking score obtained on NADPH oxidase corresponds to α bisabolol (∆G = −10.62 kcal/mol), followed by 5-methyl-1, 2, 3, 4-tetrathiane (∆G = −9.33 kcal/mol). In silico analysis of eugenol against the β-glucosidase effector protein of *Fusarium solani* f. sp. *piperis* revealed that eugenol showed promising fungicidal activity and cytotoxic activity similar to that of tebuconazole fungicide. β-glucosidase showed good binding interaction with eugenol by forming amino acid residues with Arg177 followed by a hydrogen bond with Glu596, indicating an essential role in the interactions and justifying the antifungal action of this compound [42].

Out of the six chemically synthesized compounds evaluated, oxalic acid, salicylic acid, azoxystrobin, and curzate showed good binding interactions with the effector proteins of *P. cubensis*. Likewise, the resistance mechanisms of QoI fungicides (azoxystrobin) were studied earlier through molecular docking studies of the cytochrome b gene of *Peronophythora litchi,* the causal agent of litchi downy mildew [43]. They revealed that QoI fungicides (azoxystrobin) are potent inhibitors of *P. litchi.* Similarly, it was mentioned that salicylic acid has antifungal and antibacterial activity. They conducted homology modeling and docking analysis of salicylic acid against the PR1 protein of *Xanthomonas oryzae.* The results showed that salicylic acid has more binding affinity and interaction with the PR1 protein [44]. Among the five botanicals tested, garlic bulb extract showed maximum inhibition (71.42%) followed by clove oil (64.51%). Garlic bulb extract at a 15 percent concentration showed maximum inhibition of sporangial germination (71.42%), followed by clove oil at a 5 percent concentration (71.76%). Results from earlier reports found that the concentrations of 50–1000 μg ml/1 allicin in garlic juice reduced the severity of cucumber downy mildew caused by *P. cubensis* by approximately 50–100 per cent under controlled conditions [42]. The volatile antimicrobial substance allicin (dially thiosulphinate) from garlic (*Allium sativum*) at concentrations 50–100 μg/mL reduced the severity of *P. cubensis* on cucumber by approximately 50–100% [45]. In addition, clove oil at 4 percent effectively reduced the downy mildew incidence in cucumber [46].

## 5. Conclusions

The phytochemical compounds belonging to glucosyl flavones, terpenoids and flavonoids have shown good binding interactions on the ONE4 effector protein of *P. cubensis*. Among the 15 antimicrobial compounds from botanicals tested, allicin (−7.5 kcal/mol), allixin (−7.5 kcal/mol), allyl acetate (−7.2 kcal/mol), alliin (−5.9 kcal/mol), eugenol acetate (−5.5 kcal/mol), and (E)-β-caryophyllene (−5.5 kcal/mol) showed the highest binding affinities, and salicylic acid (−12.1 kcal/mol), oxalic acid (−11.2 kcal/mol), curzate (−7.7 kcal/mol) and azoxystrobin (−6.6 kcal/mol) are the chemical compounds which exhibited the highest binding affinities. Among the five botanicals tested, garlic bulb extract showed maximum inhibition (71.42%), followed by clove oil (64.51%). However, it is important to evaluate the phytochemicals and chemically synthesized compounds under in vitro and in vivo conditions and botanicals under in vivo conditions to validate the prediction studies as many phytochemicals and chemically synthesized compounds have a potential role in the inhibition of *P. cubensis* in cucumber.

## Figures and Tables

**Figure 1 plants-12-02202-f001:**
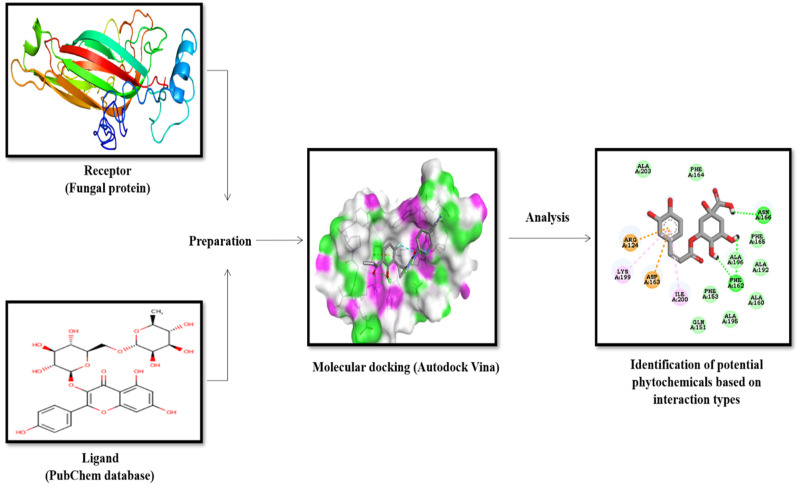
The workflow of molecular docking analysis of phytochemicals, antimicrobial compounds, and chemically synthesized compound agents with proteins of *P. cubensis*.

**Figure 7 plants-12-02202-f007:**
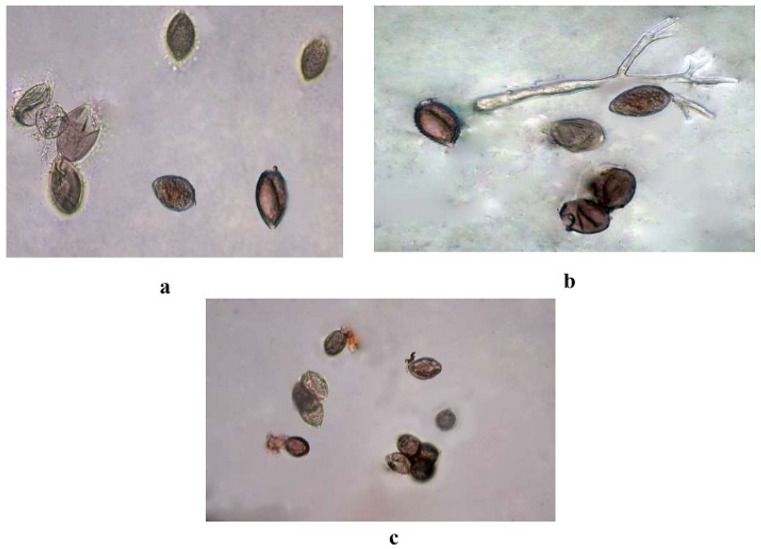
Inhibition of sporangia germination (**a**) clove oil @5% (**b**) Garlic @15% (**c**) Control.

**Table 2 plants-12-02202-t002:** List of ligands such as terpenoids, glucosyl flavones, flavonoids, megastigmane derivatives, indolic secondary metabolites, flavone glucosides, polyphenols, antimicrobial compounds, and chemically synthesized compounds used for molecular docking analysis.

Group	Sl. No.	Compounds	PubChem/Drug Bank ID	Source
Terpenoids	1	Cucurbitacin-A	5281315	*Cucumis sativus* L.
	2	Cucurbitacin-B	5281316	*Cucumis sativus* L.
	3	Cucurbitacin-C	5281317	*Cucumis sativus* L.
	4	Cucurbitacin-D	5281318	*Cucumis sativus* L.
	5	Cucurbitacin-E	5281319	*Cucumis sativus* L.
	6	Cucurbitacin-I	5281321	*Cucumis sativus* L.
Glucosyl flavones	7	Cucumerin-A	44257649	*Cucumis sativus* L.
	8	Cucumerin-B	44257648	*Cucumis sativus* L.
Flavonoids	9	Vitexin	5280441	*Cucumis sativus* L.
	10	Isovitexin	162350	*Cucumis sativus* L.
	11	Orientin	5281675	*Cucumis sativus* L.
	12	Isoorientin	114776	*Cucumis sativus* L.
Megastigmane derivatives	13	Cucumegastigmane-I	16105430	*Cucumis sativus* L.
	14	Cucumegastigmane-II	16105434	*Cucumis sativus* L.
	15	(+)-Dehydrovomifoliol	688492	*Cucumis sativus* L.
Indolic secondary metabolites	16	Indole-3-aldehyde	10256	*Cucumis sativus* L.
	17	Indole-3-carboxylic acid	69867	*Cucumis sativus* L.
Flavone glucosides	18	Isoscoparin	442611	*Cucumis sativus* L.
	19	Saponarin	441381	*Cucumis sativus* L.
	20	Vicenin-2	442664	*Cucumis sativus* L.
	21	Apigenin-7-*O*-glucoside	5280746	*Cucumis sativus* L.
	22	Quercetin-3-*O*-glucoside	5280804	*Cucumis sativus* L.
	23	Isorhamnetin-3-*O*-glucoside	5318645	*Cucumis sativus* L.
	24	Kaemferol-3-*O*-rhamnoside	5316673	*Cucumis sativus* L.
Polyphenol	25	4-hydroxycinnamic acid	637542	*Cucumis sativus* L.
Antimicrobial compounds	26	Carrageenan	71597331	*Acanthophora specifira* V.
	27	Acyclovir	135398513	Chemically synthesized
	28	5-Azacytidine	9444	Chemically synthesized
	29	Cytarabine	6253	Chemically synthesized
	30	Ribavirin	37542	Chemically synthesized
	31	Ridovudine	35370	Chemically synthesized
	32	Ningnanmycin	44588235	*Streptomyces noursei* var. *xichangensis*
	33	Vidarabine	21704	Chemically synthesized
	34	Acycloguanosine	135398513	Chemically synthesized
	35	2-Thiouracil	1269845	Chemically synthesized
	36	Moroxydine hydrochloride	76621	Chemically synthesized
	37	Luotonin A	10334120	*Peganumnigella strum* B.
	38	Tylophorinine	264751	*Cynanchum, Pergularia* and *Tylophora*
	39	Antofine	639288	*Cynanchum komarovii* I.
	40	Deoxytylophorinine	6426880	*Cynanchum komarovii* I.
	41	Pyrroloisoquinoline	86733878	*Cynanchum komarovii* I.
	42	Pulmonarin-A	76335702	*Synoicum pulmonaria*
	43	Pulmonarin-B	76313965	*Synoicum pulmonaria*
	44	Streptindole	135431	*Streptococcus faecium*
	45	Tryptanthrin	73549	*Indigofera tinctoria* L.
	46	Essramycin	24829329	*Streptomyces sp.*
	47	Chlorogenic acid	1794427	*Solanum tuberosum* L.
	48	Peonidin	441773	*Solanum tuberosum* L.
	49	Swertianolin	5858086	*Swertia chirayita* L., *S. macrosperma* L., *Gentiana campestris* L.
	50	Zidovudine	35370	Chemically synthesized
Clove	51	Eugenol	3314	*Syzygium aromaticum*
	52	Eugenol acetate	7136	*Syzygium aromaticum*
	53	(*E*)-β-Caryophyllene	5281515	*Syzygium aromaticum*
Garlic	54	Allyl acetate	11584	*Allium sativum*
	55	Allicin	65036	*Allium sativum*
	56	Allixin	86374	*Allium sativum*
	57	Alliin	87310	*Allium sativum*
Neem	58	Azadiractin a	5281303	*Azadirachta indica*
	59	Nibolin b	6443005	*Azadirachta indica*
	60	Azadiractin b	16126804	*Azadirachta indica*
	61	Nimbin	108058	*Azadirachta indica*
Tulasi	62	Gallic acid	370	*Ocimum tenuiflorum*
	63	Catechol	289	*Ocimum tenuiflorum*
	64	Cinnamic acid	444539	*Ocimum tenuiflorum*
Pudina	65	Menthol	1254	*Mentha spicata* subsp. *spicata*
Chemically synthesized compounds	66	Azoxystrobin	3034285	Chemically synthesized
	67	Ridomil	3036793	Chemically synthesized
	68	Kresoxim methyl	6112114	Chemically synthesized
	69	Curzate	5364079	Chemically synthesized
	70	Oxalic acid	971	Chemically synthesized
	71	Salicylic acid	338	Chemically synthesized

**Table 3 plants-12-02202-t003:** Linear combination of two structural descriptors for model quality assessment.

Sl. No.	Protein	Template	Query Coverage (%)	Per Cent Similarities (%)	GMQE	QMEAN
1	Cytochrome oxidase subunit 1	7 jro 1. B	99	99.0	0.77	0.67
2	QNE 4	5 gnc 1. A	100	93.94	0.16	0.43 +/− 0.05

**Table 4 plants-12-02202-t004:** Calculated secondary structures (in percentage) by SOPMA.

Secondary Structures	QNE4	Cytochrome Oxidase Subunit 1
Alpha helix %	42.36	44.77
Extended strand %	12.38	21.17
Beta turn %	8.70	8.27
Random coil %	36.56	25.79

**Table 12 plants-12-02202-t012:** Evaluation of botanicals against downy mildew under in vitro conditions.

Treatment	% Inhibition of Sporangial Germination		
	Concentration		
	5%	10%	15%
Clove	47.41 (6.95) **	57.14 (7.62)	64.51 (8.09)
Garlic	57.14 (7.62)	61.9 (7.93)	71.42 (8.51)
Tulsi	38.09 (6.25)	47.61 (6.97)	52.38 (7.30)
Pudina	38.09 (6.25)	47.41 (6.95)	57.14 (7.62)
Neem	33.33 (5.85)	42.85 (6.62)	57.14 (7.62)
Control	16 (4.12)	16 (4.12)	16 (4.12)
Mean	42.11 (6.45)	46.43 (6.78)	51.91 (7.15)
	Treatment	Concentration	Treatment X Concentration
S.Em±	0.271	0.177	0.469
CD (*p* < 0.05)	0.776	0.508	1.344

Figures in parentheses are without transformed values. ** Values in bracket are arc-sin transformed values.

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
