# Peer review of "Identification of Potential Phytochemical/Antimicrobial Agents against *Pseudoperonospora cubensis* Causing Downy Mildew in Cucumber through In-Silico Docking"

_plants, 2023, doi:10.3390/plants12112202_

Round 1

Reviewer 1 Report (Previous Reviewer 4)

Manuscript plants-2326195 presents an in-silico assessment of antifungal agents, from cucumber, from other botanical sources, and synthetics, on two protein targets of Pseudoperonospora cubensis. There are several problems with the manuscript and publication cannot be recommended.

Much of the homology modeling methods and results are outside the scope of most readers of Plants. It would be better to place this information as supplementary materials. Figure 2: These structures do not really add anything to the manuscript and can be deleted.

Table 2: It would be better to either include the molecular structures in the table or provide a figure with the molecular structures. The subsection “Antioxidant” is misleading; all of the compounds are antioxidant to some degree. Maybe better to label these two compounds as flavonoids or glucosyl flavones. Entry 19: Is this name correct? Maybe Saponarin? Entries 21-24: The "o" should be capitalized and italicized. Entry 26: Carrageenan [correct the spelling]. Entry 53: This must be (E)-β-Caryophyllene [correct the name and spelling]. Garlic subsection: Is there a reason ajoene was not included? Entry 56: Should this be Allixin? Entry 57: Should this be Alliin?

Table 8: Many of the structures shown in Figures 5-11 do not agree with the compounds listed.

Table 10: Some of the structures are difficult to see. Maybe better to have all atoms shown in black rather than colors. Entry 8: The name should be Alliin; the structure is incorrect. Entry 10: This should be (E)-β-caryophyllene. The structure (and subsequent docking) is incorrect. There are no hydrogen bond donors or acceptors in (E)-β-caryophyllene (or caryophyllane), so hydrogen bonding to the protein is not possible.

Figure 6A: The structure shown is not oxalic acid. Figure 6I: Alliin? The structure is not correct.

Figure 7B: This is not oxalic acid. Figure 7E: The structure looks like allicin again. Figure 7G: The structure shown does not look like the structure in Table 10. Figure 7H: Alliin? The structure is not correct. Figure 7I: The structure looks like eugenol.

Figure 10A: The structure shown is cinnamic acid. Figure 10B: Allicin. Figure 10F: The structure shown is eugenol. Figure 10G: The structure shown does not look like the structure in Table 10. Figure 10I: The structure shown is allixin again.

Figure 11B: This is not oxalic acid; it is eugenyl acetate. Figure 11C: The structure looks like azoxystrobin. Figure 11E: This is allicin again. Figure 11G: The structure shown does not look like the structure in Table 10. Figure 11H: This is allixin. Figure 11I: This is allixin again.

Since the many of the structures in the docking are incorrect, the docking scores are not reliable, and the discussion is meaningless at this point. Line 463: The structures shown are not oxalic acid, so the interacting residues are incorrect. Line 472: Caryophyllane has no hydrogen bond donors or acceptors.

Table 12, entry 9: The structure is incorrect.

Table 13, Clove: These % inhibition values look to be reversed. Percent inhibition should increase with increasing concentration.

My recommendation is to:

1.       Double-check all molecular structures and include a figure with all of the structures drawn, including stereochemistry.

2.       Re-do the docking with all of the structures and re-evaluate the docking scores and intermolecular interactions.

3.       Present the docking scores and the interacting residues in one table.

Other editorial corrections needed:

Enzymes, chemical compounds, viruses, common names of plants, are not proper nouns and should not be capitalized.

The references are not formatted correctly for Plants. The order of the manuscript sections is not correct for Plants.

There are numerous spelling errors. A technical editor should proof-read the penultimate manuscript.

Line 157: A0 must be Å.

Line 179: filtration

Line 219: The formula is poor resolution; better to use the "insert equation" of WORD.

Line 224: Should 95% actually be 5%?

Line 433: This should be (E)-β-caryophyllene.

Author Response

Please find the attachment

Reviewer 2 Report (Previous Reviewer 2)

The text bellow contains comments on manuscript entitled “Identification of potential phytochemical/antimicrobial agents against Pseudoperonospora cubensis causing downy mildew in cucumber through In-silico docking” given for revision after resubmission.

Page 1, Line 15: Is it photochemical or phytochemical?

I think that section 2.3.2 should be written in a more compact way. It is more important to give information about the rpm/rcf values of centrifugation and the model of the centrifuge, rather than its dimensions.

The chemical structures in table 9, 10, 11 and 12 should be given in a better quality.

I highly advice the authors check once again the text for grammar and technical inconsistencies in the text and tables.

Author Response

Please find the attachment

Reviewer 3 Report (Previous Reviewer 1)

The paper has been corrected significantly and I think that the final version of this paper can be considered for publication.

Round 2

Reviewer 1 Report (Previous Reviewer 4)

This manuscript is much improved over the previous version. There are, however, minor corrections needed:

Line 38: “pickling” should not be capitalized.

Line 50: “…tons of cucumber were produced…”

Line 55: “watermelon” should not be capitalized.

Line 71: “Now a day’s” is not correct; replace with “nowadays”.

Line 85: “…in this study, i.e.,… [add commas].

Line 87: “…used against many…” [delete the comma].

Line 88: “L.” not italicized.

Table 2, entry 19: Saponarin [correct the spelling]

Table 2, entry 65: “subsp.” not italicized.

Line 139: “Gasteiger” should be capitalized.

Table 7, entry 19: Saponarin [correct the spelling]

Table 7, entry 53: (E)-β-caryophyllene [correct the spelling].

Figures 4-7: These structures are impossible to visualize. I suggest putting each structure on a separate page and offer them as supplementary material.

Figure 5 legend: Saponarin [correct the spelling].

Figures 8 and 9: These are too small to see. Please enlarge them (comparable in size to Figures 10 and 11).

Figure 9 legend: Saponarin [correct the spelling].

Line 397” “indole-3-aldehyde” [not capitalized].

Line 400: “isocarpin” [not capitalized].

Line 499: saponarin [correct the spelling].

Table 10, entry 2: Saponarin [correct the spelling]. Note the structure shown is Saponarin.

Line 518” “good inhibitory action” is misleading. The compounds were not screened for inhibitory activity. Maybe better to say "good docking scores" or "good binding interactions".

Line 523: (E)-β-caryophyllene [correct the spelling].

Line 546: “salicylic acid” [not capitalized].

Line 560: “excellent inhibitory action” is misleading. The individual compounds were not screened for inhibitory activity. Maybe better to say "good docking scores" or "good binding interactions".

Line 563: (E)-β-caryophyllene [correct the spelling].

Author Response

Please find the revised version as well as the response to reviewers

This manuscript is a resubmission of an earlier submission. The following is a list of the peer review reports and author responses from that submission.

Round 1

Reviewer 1 Report

Manuscript ID: plants-2135565

Title: Identification of potential phytochemical/antimicrobial agents against Pseudopernospora cubensis causing downy mildew in cucumber through in-silico docking

Generally, presented approach to identify ligands from botanical and chemical sources which have a potential role in the inhibition of P. cubensis in cucumber is very interesting.

Although, the results are promising, some details should be improved and explained. I would like to make some comments that authors could take into account to improve the overall quality of the manuscript.

Comments:

Line 42: The medical properties of cucumber should be support by much more current literature. Using only one literature position (published 1927) to support mentioned medical properties is not sufficient.

Line 49/50: The data from FAO Statistical Yearbook – World Food and Agriculture (2021) were not properly interpreted, for example – look to FAO Statistical Yearbook: “World vegetables production grew faster between 2000 and 2019, as it went up 65 percent, or 446 million tonnes, to 1 128 million tonnes in 2019. The five main vegetable species accounted for 42–45 percent of the total during the period: tomatoes (16 percent in 2019), onions (9 percent), cucumbers and gherkins (8 percent), cabbages (6 percent) and eggplants (5 percent).” – it means cucumbers and gherkins production in 2019 was 1 128 million tonnes x 8% = 90.24 million tonnes. More current data (2021) you can find using https://www.fao.org/faostat/en/#search/cucumber

Line 91: Tables should be numbered consistently and presented in the same order in the manuscript, however Table 2 was presented on page 3 and next Table 1 on page 8.

Line 170: This paragraph should contain own subtitle because does not describe “Methanol extraction method”.

Figures and structures: Figure 1 – low resolution, probable some arrows and text were masked by black background; Figure 3 – low resolution of text below plots; Figures 4, 5 and 8-11 – very low resolution of structures.

Line 189: The software OPSTAT was used, the author (Prof. O.P. Sheoran) asked “For citations please use “Sheoran, O.P; Tonk, D.S; Kaushik, L.S; Hasija, R.C and Pannu, R.S (1998). Statistical Software Package for Agricultural Research Workers. Recent Advances in information theory, Statistics & Computer Applications by D.S. Hooda & R.C. Hasija Department of Mathematics Statistics, CCS HAU, Hisar (139-143)”.

Line 187: maybe one more subtitle?

Line 187: It was mentioned that ANOVA was done but I do not see results.

Reviewer 2 Report

The text bellow contains comments on manuscript entitled “Identification of potential phytochemical/antimicrobial agents against Pseudopernospora cubensis causing downy mildew in cucumber through In-silico docking”

The manuscript is focused on homology modeling and in silico docking analysis of 50 phytochemicals from cucumber, 15 antimicrobial compounds from 18 botanical sources and six compounds from chemical sources against two effector proteins of 19 Pseudoperonopsora cubensis linked to cucumber downy mildew.

To my opinion the English language and grammar should be improved. All the tables must be better arranged. All figures must be replaced with a higher quality once.

Abstract: Cucumis sativus and P. cubensis must be italics and this is valid for the whole text. P. cubensis (Pseudopernospora cubensis) must be mentioned with its full name before to use the abbreviation.

Reviewer 3 Report

Jhansi Rani Nagaraj and her co-authors wrote a interesting manuscript about in silico docking/screening of potential agents against Pseudoperonospora cubensis, an oomycete causing downy mildew on cucurbits - an economically important disease.

The authors have chosen two P. cubensis proteins for this purpose: cytochrome C oxidase subunit I and the effector protein PcQNE-4. Unfortunately, the authors did not describe why they have choosen these two proteins in particular. The manuscript unfortunately also lacks a general introduction on immunity of plants in general and in particular to oomycetes. Here one could classify these two proteins, if they play a role in the immune response at all. The authors should care about Salicylic acid dependend and independed responses, on answers on pathogen-associated molecular pattern, on general and specific immune answers. They can compare the knowledge about the system cucumber / P. cubensis with the huge knowlege about potato and Phytophthora infestans, the oomycete causing late blight.

The authors' idea was to find substances that interact with these two proteins. So they searched the literature and selected known ingredients from cucumber (Cucumis sativus L.), garlic (Allium sativum
L.), onin (Allium cepa L.), clove (Syzygium aromaticum (L.) Merr. & L. M. Perry), tulsi (Ocimum tenuiflorum L.), wild mint (Mentha arvensis L.) and neem (Azadirachta indica A. Juss.) for their study. They also added some synthesized compounds.

A total of 71 substances were selected. Unfortunately, it is not clear why these plants were used as a source. It is also not clear why it was limited to these 71 substances. A quick reaxis search shows that over 340 substances have already been described just from cucumber alone.

The substances were classified/grouped in table 2, whereby this classification is not very logical: for instance terpenoid vs antioxident - a classifier for a basic chemical structure vs. a classifier for a chemical property. Cucumerins are flavons by the way.

The amino acid sequence of cytochrome c oxidase subunit I in table 1 is wrong. One amino acid is missing. Please compare to: https://www.ncbi.nlm.nih.gov/protein/AEA38564.1

I really hope, this is only a mistake during the creation of the manuscript and not a mistake of the whole docking study.

The authors wrote in their abstract (line 19ff): "against  two  effector  proteins  of Pseudoperonopsora cubensis linked to cucumber downy mildew". Is cytochrome c oxidase an effector protein?
line 23 ff: "The results of the molecular docking analysis showed that various effector proteins of P. cubensis showed good binding affinities with [...]" various? Here, in this manuscript only ONE effector protein was analysed. Miaoying Tian at al. for instance could found a series of effector proteins in P. cubensis (DOI:10.1094/MPMI-08-10-0185)

The main part of this study was virtual docking. 1st step was 3D optimisation (force field) of the ligands. No care was taken about the protonation state. The authors just have taken the neutral state.

No scoring function was given for the docking evaluation. A binding affinity less than -6  kcal/mol was used: not clear whether enthalpy and entropy is meant. The given reference (Klebe 2006) does not tell anything about a threshold, specially not about the 6 kcal/mol.

Table 6 reads: Dock score for binding affinity (Kcal/mol) of viral coat proteins
why viral? (please use a small k for kcal)

The authors need to show, that the binding of the ligands to the proteins are specific and not unspecific. A critical plausibility check of the received docking results is missing. What is known from literature about flavonoids or polyphenols and proteins?

A last part of the manuscript is the in vitro evaluation of "botanicals". Therefor two kinds of extracts were generated (water and methanol) and evaluated for P. cubensis sporangial germination.

This whole part is very non-scientific. i would just delete it.

Line 149 (v./v.) <--- dont you mean mass / volume?

line 153: "A hundred grams of plant material" no plant source was given
line 153f: dried in a hot air oven (80 °C) <-- you can be sure: all high volatile substances are gone or destroyed
line 155:  was extracted in distilled water for 6 h at slow heat <-- slow heat is not really specific
line 157: centrifuged at 5000 rpm  <-- what is the diameter of the centrifuge? or the centrifugal force (how many g)
line 158: autoclaved at 121 °C <-- if not destroyed at 80°C, than now. Why not steril filtrated?
line 158: required concentrations (5, 10 and 15%) were prepared  <-- concentration? dry mass / volume?

line 170: new sub-headline is needed

line 170ff: fresh sporangia from infected leaves <-- how the leaves where infected? artificially or naturally? Which strain of P. cubensis? a mixture? or definded?
line 171:  to make a sporangial suspension. <-- which concentration? was it checked by microscope?
line 177: why multiplied by 10

line 187: The experiment was laid out with a Completely Randomized Design (CRD) with three replications.
explain your CRD

line 479ff:  The volatile antimicrobial substance allicin (diallythiosulphinate)  from  garlic  (Allium  sativum)  at  concentrations  50-100  μg/ml  
reduced  the  severity  of  P.  cubensis  on  cucumber  by  approximately  50-100  %  (Chen  et al.,2016)
<--- it is not clear where the result comes from: own or from Chen et al.
Be sure, in your own extracts there is no allicin anymore. allicin is very unstable

line 476: vitro was carried out at different concentrations of five botanicals. [...] (Table13).
I count 6 botanicals in table 13

Table 13: i dont understand "Treatment mean", makes no sence to mean the results of different concentration

i dont understand the values with and without brackets

where does the values from "Castor", "Pudina", and "Ginger" come from? Not mentioned in 2.3

The manuscript is written in a good, but improvablen English. It does NOT fullfill all the criteria of a scientific paper. Not all procedures are described in such a way that they are comprehensible. Many figures have low quality, and may be moved to the ESI. References are missing in many cases. The topic of the manuscipt is very interesting, but the authors could not the authors could not convince with their results.

There are many spelling mistakes, starting with Pseudoper*o*nospora in the headline.

I suggest to the editor to REJECT this manuscript

Reviewer 4 Report

In manuscript plants-2135565, the authors have prepared two protein targets of Pseudoperonospora cubensis using homology modeling. A total of 71 compounds were then used for molecular docking with the two protein structures. There are several shortcomings in the manuscript and publication cannot be recommended.

1.       What was the basis for selecting the two proteins? Have inhibition studies on these two protein targets been carried out? Why not screen against all of the proteins available to check for target selectivity?

2.       How were the compounds in Table 2 selected? Are they known to exhibit antifungal activities? Maybe this information can be included in this table (e.g., fungal species and MIC or IC50 values). What was the basis for selecting the “Antimicrobial compounds”? Several of these are antivirals. It would be better to use known antifungals to serve as “positive controls”.

3.       The figures (Figure 1, Figure 2, Figures 4-11), equations (Equation 1, line 183), and structures (Allicin) are poor (poor resolution, too small, or difficult to see).

4.       Why was the cytochrome oxidase 2 subunit selected? The protein is a component of a transmembrane protein complex. Interactions of small-molecule ligands with only one subunit may not explain any inhibitions of the protein complex.

5.       The in-vitro evaluation used extracts rather than the compounds listed in Table 2, so comparison between activity and docking scores is not possible.

6.       Table 13 is confusing. For clove oil, are these values in the correct order (i.e., the % inhibition actually decreases with increasing concentration)? What are the values in parentheses, standard deviations? What does the asterisk mean? What does the treatment mean tell you? It would be better to determine the IC50 values from the three determinations.

7.       There are numerous errors in spelling, punctuation, use of italics, capitalization, etc.; the manuscript should be carefully proofread by a technical editor.

My recommendation is to expand the scope of the study by including all potential protein targets of Pseudoperonospora cubensis , focusing on known antifungal phytochemicals and synthetic antifungal drugs for in silico screening, and in vitro screening of promising in silico hits with available compounds.